# Effects of Plant-Derived Glycerol Monolaurate (GML) Additive on the Antioxidant Capacity, Anti-Inflammatory Ability, Muscle Nutritional Value, and Intestinal Flora of Hybrid Grouper (*Epinephelus fuscoguttatus*♀ × *Epinephelus lanceolatus*♂)

**DOI:** 10.3390/metabo12111089

**Published:** 2022-11-10

**Authors:** Xuehe Li, Yuanming Yi, Jiahua Wu, Qihui Yang, Beiping Tan, Shuyan Chi

**Affiliations:** 1College of Fisheries, Guangdong Ocean University, Zhanjiang 524088, China; 2Aquatic Animals Precision Nutrition and High Efficiency Feed Engineering Research Center of Guangdong Province, Zhanjiang 524088, China; 3Guangdong Provincial Key Laboratory of Aquatic Animal Disease Control and Healthy Culture, Zhanjiang 524088, China

**Keywords:** hybrid grouper, glycerol monolaurate, plant-derived additives, anti-inflammatory ability, antioxidant capacity, muscle nutritional value, intestinal flora

## Abstract

In a context where the search for plant-derived additives is a hot topic, glycerol monolaurate (GML) was chosen as our subject to study its effect on grouper (*Epinephelus fuscoguttatus*♀ × *Epinephelus lanceolatus*♂). Seven gradient levels of GML (0, 600, 1200, 1800, 2400, 3000, and 3600 mg/kg) were used for the experiment. Based on our experiments, 1800 mg/kg GML significantly increased the final body weight (FBW) and weight gain rate (WGR). GML increased the activity of superoxide dismutase (SOD) and glutathione peroxidase (GSH-Px) and decreased malondialdehyde (MDA). Adding 1800 mg/kg GML also significantly increased the levels of lauric acid (C12:0) (LA), n-3 polyunsaturated fatty acids (PFA), and the n-6 PFA-to-n-3/n-6 ratio, while significantly decreasing the levels of saturated fatty acids (SFA). Dietary supplementation with GML significantly inhibited the expression of pro-inflammatory factors and reduced the occurrence of inflammation. GML improved intestinal flora and the abundance of beneficial bacteria (*Bacillus*, *Psychrobacter*, *Acinetobacter*, *Acinetobacter*, *Stenotrophomonas*, and *Glutamicibacter*). It provides a theoretical basis for the application of GML in aquafeed and greatly enhances the possibility of using GML in aquafeed. Based on the above experimental results, the optimum level of GML in grouper feed is 1800 mg/kg.

## 1. Introduction

Hybrid grouper (*Epinephelus fuscoguttatus*♀ × *Epinephelus lanceolatus*♂) has become one of the most important marine economic fish in China because of its tender meat and delicious taste. In recent years, the production of grouper culture has been rising, and the scale of culture has been expanding. However, the rapid development of the intensive culture of grouper may induce oxidative stress in fish, caused by aquatic animals, and affect the antioxidant protection capacity of aquatic animals [1].

Oxidative stress is caused by the inordinate generation of free radicals and reactive oxygen species (ROS), resulting in a reduction in the body’s antioxidant defenses [2,3]. It causes biomolecular damage and metabolic and physiological dysregulation, which in turn results in an imbalance in the animal’s immune function and inflammatory response [4,5]. Oxidative stress promotes multiple transcription factors that allow for a high expression of pathways associated with inflammation [6]. Inflammation caused by oxidative stress is the etiology of many chronic diseases [7]. Inflammatory diseases can reduce aquatic animal productivity, reproductive performance, and disease resistance, causing huge economic losses not only to farmers but also to consumers [8].

In the past several years, antibiotic treatments have been one of the main treatments to counteract oxidative stress and inflammatory diseases and improve disease resistance in animals [9,10]. However, antibiotics have had a serious impact on sustainable development, human health, and the ecological environment due to drug resistance and residues [11,12]. The European Union (EU) has imposed a ban on the application of antibiotics as additives in animal feed since 2006 [13]. Therefore, further development and research on additives with the potential to replace antibiotics has become a pressing matter of urgency. Plant extracts are recognized as a promising source of antioxidants and anti-inflammatory substances that have been confirmed and suggested for combatting diseases associated with oxidative stress [14].

Glycerol monolaurate (GML) is a naturally occurring compound commonly found in coconut oil and American spice and is a food additive of plant origin permitted by the US Food and Drug Administration (FDA) [15,16,17]. It has excellent antibacterial, antiviral, antioxidant, and emulsifying properties [18,19]. Because GML is a natural compound, it does not cause either resistance in animals or drug residues. Recently, it has been shown that GML has a beneficial effect on terrestrial animals’ (piglets [20], broilers [21,22], laying hens [23], weaned lambs [24]) growth and development, significantly promoting growth, improving antioxidant capacity, and reducing the inflammatory response. However, there is a huge gap in the study of GML in aquatic animals; the main ones known are *Litopenaeus vannamei* [25], *Danio rerio* [26], and *Pelodiscus sinensis* [27]. The aim of this paper is to explore the effects of GML on antioxidant capacity, disease resistance, and inflammatory response in grouper to improve the feasibility of GML application in aquatic animals and to provide a theoretical basis for the further substitution of GML for antibiotics.

## 2. Materials and Methods 

This study was reviewed and approved by the Institutional Animal Care and Use Committee (IACUC) of the College of Aquatic Sciences, Guangdong Ocean University. The approval code is GDOU-IACUC-2021-A2013 and the approval date was 11 April 2021.

### 2.1. Experimental Design and Diets Preparation

This experiment followed the method of the Association of Official Analytical Chemists (AOAC) [28] to prepare a basal diet. Seven iso-nitrogenous and iso-lipidic diets were formulated according to Table 1. The steps for making the feed refer to our previous production process [29].

### 2.2. Experimental Animals and Breeding Management

Juvenile grouper were sourced from the Yongsheng Fish Hatchery (Zhanjiang, China). Before the start of the experiment, all the grouper were placed in several fiberglass barrels (2 m^3^) for a fortnight to allow them to acclimatize to the culture conditions and reach the target weight. Feeding was stopped 24 h before the start of the experiment. A total of 840 healthy and active fish (9.09 ± 0.01 g) were chosen and divided into 7 groups of 4 replicates of 30 fish per tank for an 8-week trial. All fish were fed twice daily on a full stomach (7:30, 17:30). The water environment needed to be sustained at the following levels: temperature of 28–30 °C, salinity of 28–30, pH 7.5–8.0, and dissolved oxygen levels of ≥5.0 mg/L.

### 2.3. Sample Collection

After the breeding experiment, groupers were starved for 24 h before samples were collected [30]. All fish were anesthetized with MS-222 before the sample collection. Four fish in each tank were chosen freely to collect their livers and then transferred to −80 °C after being temporarily frozen in liquid nitrogen until analysis. Four fish were randomly selected from each tank, and the head kidneys were taken, placed in RNA later at 4 °C 24 h, and then kept at −80 °C. The above sample collection procedures were conducted on ice and were based on previous experimental methods [31,32].

### 2.4. Liver Antioxidant Enzyme Activity Analysis 

The activities of catalase (CAT), superoxide dismutase (SOD), glutathione peroxidase (GSH-Px), malondialdehyde (MDA), and lysozyme (LZM) were analyzed using commercial ELISA kits (Shanghai Enzyme-linked Biotechnology Co., Ltd., Shanghai, China). Liver samples were homogenized into 10% homogenate (EasyWeLL Series JY98-IIIN Model Cell Disruptor) with iced saline (sterile) at a ratio of 1:9. After cryogenic homogenization (2500 rpm, 10 min), the supernatant was extracted and then analyzed according to the instructions. The above measurements of enzyme activity were performed on ice. 

### 2.5. Gene Expression Analysis 

The head kidney total mRNA was obtained using an RNA extraction kit (TansGen Biotech, Co., Ltd. (Beijing, China)). The complementary DNA (cDNA) was transcribed using an Evo M-MLV RT kit (Accurate Biotechnology (Accurate Biotechnology Co., Ltd. (Changsha, China)). 

The gene expressions of toll-like receptor 2 (*TLR2*), myeloid differentiation primary response gene 88 (*myd88*), interleukin 1β *(IL-1β*), interleukin 10 *(IL10*), interleukin 8 (*IL8*) were performed on Roche Light Cycler 480Ⅱ (Switzerland) using an SYBR^®^ Green Pro Taq HS Premix II kit (Accurate Biotechnology Co., Ltd. (Changsha, China)). The primer sequences of genes are shown in Table 2. The results of real-time qPCR were analyzed by previous authors [33].

### 2.6. Vibrio parahaemolyticus Challenge Test

The *Vibrio parahaemolyticus* used for the bacterial challenge test was provided by the Key Laboratory of Pathogenesis Biology and Epidemiology of Aquatic Economic Animals of Guangdong Province. The strains were taken out from −80 °C and thawed and shaken well, and then 200 µL were taken and inoculated into Luria–Bertani (LB) solid medium for rejuvenation of the strain (37 °C, 24 h). Single colonies with strong growth were added to LB liquid medium and incubated at 37 °C for 18–24 h. At the end of the culture, the bacteria were collected by centrifugation at 4 °C (4000 rpm, 10 min), washed with sterile phosphate-buffered saline (PBS), and then centrifuged again. This process was repeated 2–3 times until the solution was clarified. Different volumes of PBS were added for dilution until OD = 1 (10^9^ cfu/mL). Then the solution was diluted to 10^8^, 10^7^, and 10^6^ for the pre-test. The results show that the median lethal dose (LD50) was 2 × 10^7^ cfu/mL. Ten grouper were collected randomly from each tank and injected with 200 μL of the bacterial solution slowly into the abdominal cavity for a 7 d bacterial challenge test. At the end of the 7 d challenge test, four livers were collected from each vessel to detect immune and antioxidant-related enzyme activities.

### 2.7. Statistical Analysis

Statistical analysis of the experimental data was conducted using SPSS 17.0 software. One-way ANOVA was used to analyze the data, and Duncan’s multiple comparisons were applied if there was a significant difference (*p* < 0.05). The experimental data were expressed as means ± standard deviation (X ± SD). 

## 3. Results

### 3.1. Effects of Dietary GML Levels on the Antioxidant Index in Liver and Serum for Grouper

FBW and WGR increased and then decreased with the addition of GML, and both had a maximal value in the G1800 group (*p* < 0.05). However, there was no significant effect of SR between the groups (*p* > 0.05) (Figure 1).

### 3.2. Effects of Dietary GML Levels on the Antioxidant Index in Liver and Serum for Grouper

All antioxidant indices are shown in Table 3. In the serum, the group supplemented with GML significantly reduced the MDA content (*p* < 0.05); SOD activity was significantly increased (*p* < 0.05); GSH-Px was significantly higher in the G1800, G2400, and G3000 groups than in the control group (*p* < 0.05). In the liver, the G1800 group had the lowest values of MDA and the highest values of SOD activity (*p* < 0.05). The difference in CAT activity between the liver and serum was not statistically significant (*p* > 0.05). 

### 3.3. Effects of Dietary GML Levels on Muscle Fatty Acid Composition for Grouper

The composition of muscle fatty acid (FA) is shown in Table 4. The G1800 group had significantly increased levels of lauric acid (C12:0) (LA), n-3 polyunsaturated fatty acids (PUFA), n-6 PUFA, and the n-3/n-6 PUFA ratio and significantly decreased levels of saturated fatty acids (SFA) compared to the control group.

### 3.4. Effects of Dietary GML Levels on the Relative Expression of myd88 and IL10 in Head Kidney for Grouper

In this study, the expressions of inflammation-related genes in the head kidney were determined (Figure 2). The expression of *TLR2* in the G1200, G1800, G2400, and G3000 groups was significantly lower than that in the G0 group. The addition of GML significantly reduced the expressions of *myd88* and *IL-1β*, which was significantly higher in the control group (G0) than in the other groups (*p* < 0.05). GML significantly increased *IL10* expression, which was significantly higher in the G1800, G2400, and G3000 groups than in the G0 group (*p* < 0.05). There was no significant effect of GML on *IL8* (*p* > 0.05).

### 3.5. Vibrio parahaemolyticus Challenge Test Results

#### 3.5.1. Effects of Dietary GML Levels on Survival Rate in *Vibrio parahaemolyticus* Challenge Test 

After 7 days of the bacteria challenge test with *Vibrio parahaemolyticus*, the G1800 and G2400 groups showed the highest survival rates (*p* < 0.05) but were not significantly different from each other (*p* > 0.05) (Figure 3).

#### 3.5.2. Effects of Dietary GML Levels on Enzyme Activity of LZM in Bacterial Challenge Test 

The activity of LZM in the G600, G1800, and G2400 groups was significantly increased in the bacterial experiment test (*p* < 0.05) and had the highest value in the G1800 group (*p* < 0.05) (Figure 4A). 

#### 3.5.3. Effects of Dietary GML Levels on Enzyme Activity of SOD in *Vibrio parahaemolyticus* Challenge Test 

The SOD activities of G1200, G1800, G2400, G3000, and G3600 were significantly higher than that in the G0 group before the test (*p* < 0.05). The activities of SOD were significantly higher in the G1800, G2400, and G3600 groups than in G0 after the test (*p* < 0.05) (Figure 4B). 

#### 3.5.4. Effects of Dietary GML Levels on MDA Content in *Vibrio parahaemolyticus* Challenge Test 

The content of MDA was significantly lower in the G2400 group than in the control group before the test (*p* < 0.05). The MDA contents of the G1800, G2400, G3000, and G3600 groups did not change significantly after the test but were significantly lower than that of the control group (*p* > 0.05) (Figure 4C).

### 3.6. Effects of Dietary GML Levels on Intestinal Flora for Hybrid Grouper

#### 3.6.1. Effects of Dietary GML Levels on Relative Abundance of Species for Hybrid Grouper

As seen in Figure 5A, the top four intestinal flora of grouper at the phylum level were Proteobacteria, Firmicutes, Actinobacteria, and Bacteroidetes. The highest relative abundance of Firmicutes was found in the G1800 (12.37%) and G2400 groups (27.89%). At the genus level, G1800 (11.91%) and G2400 (27.68%) had the highest relative abundance of *Bacillus*, while *Vibrio* (2.05% and 2.595%) and *Enterovibrio* (0.13% and 0.77%) were significantly lower than the G0 group (Figure 5B).

#### 3.6.2. Effects of Dietary GML Levels on Shannon and Simpson Indices for Juvenile Grouper

There were no significant differences in the Shannon and Simpson indices among the seven groups (Figure 6) (*p* > 0.05).

#### 3.6.3. Analysis of the Association between GML and Intestinal Flora

The level of GML addition showed a significant positive correlation with *Acinetobacter*, *Acinetobacter*, *Stenotrophomonas*, and *Glutamicibacter* (*p* < 0.05), while a highly significant positive correlation was found with *Bacillus* and *Psychrobacter* (*p* < 0.01). However, the addition of GML inhibited *Photobacterium*, with a significant negative correlation (*p* < 0.05) (Figure 7).

## 4. Discussion

Our experimental results show that the addition of GML significantly improved the FBW and WGR of the grouper. Similar results have been verified in terrestrial animals [20,21,22,23,24]. However, the application of GML in aquatic animals [25,26] is still very rare and needs further extensive research.

Enzymatic and non-enzymatic systems make up the antioxidant system of an organism. The enzymatic system includes antioxidant enzymes such as SOD, CAT, and others, while the non-enzymatic system mainly includes oxidation products such as MDA, GSH, and other non-enzymatic antioxidants [34]. The antioxidant system scavenges excess ROS and superoxide anion radicals (O_2_^−^) from the body and reduces oxidative damage to the body [35]. Our experiment indicated that the addition of 1800 and 2400 mg/kg GML significantly raised the activities of SOD and GSH-Px and significantly reduced the content of MDA in the serum. The study in *Litopenaeus vannamei* demonstrated that GML significantly improved the activities of SOD and CAT [25]. A previous study showed that the SOD activity of *Procambarus clarkii* was significantly elevated because of the supplementation of GML [36]. Studies on laying hens [37], *Lateolabrax maculatus* [38], and weaned lambs [24] have shown that GML increased SOD and GSH-Px activities. Famurewa et al. showed that the addition of coconut oil (containing approximately 45% LA) significantly increased the activities of SOD, CAT, and GSH-Px in mice while reducing MDA levels [39]. The results of the above studies are similar to the present experiment, probably because medium-chain fatty acids (MCFAs) can supply energy more efficiently than carbohydrates for animals. As a typical derivative of MCFAs, GML can be added to the diet of juvenile animals for energy supply. GML can enter the mitochondria of hepatocytes freely for oxidation and is not dependent on carnitine carriers, allowing for a rapid energy supply [40]. The metabolic energy consumption of other nutrients and mitochondrial energy production are reduced, resulting in lower free radical production and less damage to the body due to oxidative stress [41]. Based on the above experimental results, the optimum level of GML in grouper feed is 1800 mg/kg.

FA composition is one of the most valuable indices of the nutritional value and quality of meat [42]. Experiments with *Trionyx sinensis* [27] and chickens [22,23,37] have shown that the addition of GML significantly reduced the SFA content. There was a tendency to reduce muscle SFA in rhododendrons by adding GML to the feed [43]. Our experimental results show that supplementation with GML (1800 mg/kg) reduced the SFA levels, which is generally consistent with the previous results described. SFA is the main cause of elevated human cholesterol (CHOL), triglycerides (TG), and low-density lipoprotein cholesterol (LDL-C) [44]. In this experiment, the addition of GML significantly reduced the SFA content, which both improved the muscle quality of groupers and reduced the harm caused by excessive SFA. Medium-chain fatty acids (MCFAs) have good antibacterial properties and can regulate the balance of the intestinal micro-ecosystem by inhibiting the proliferation of harmful bacteria [45]. GML, as a type of MFA, can be rapidly oxidized to provide energy to the organism [46]. The content of LA in muscle was positively correlated with the addition of GML, and the nutritional value was significantly increased. The n-6 PUFA and n-3 PUFA cannot be synthesized in the human body and can only be consumed from external sources; thus, they are called essential fatty acids (EFAs) [47,48]. The n-6 PUFA can be consumed in large quantities from vegetable oils, such as soybean oil. Fish (especially marine fish) and seal oils, are the main sources of n-3 PUFA intake for humans, but the intake of n-3 PUFA is severely inadequate [49]. Excessive intake of n-6 PUFA increases cellular inflammation and induces obesity, which is detrimental in humans [50]. The present experiment demonstrated that GML supplementation significantly increased the contents of the n-3 PUFA-to-n-6 PUFA ratio and improved n-3 PUFA intake and dietary balance. Supplementation of GML promotes the digestion and absorption of fat in animals due to its good emulsification properties [51]. Changes in the FA composition of grouper muscle also reflect the positive effect of GML on the nutritional quality of grouper.

Oxidative stress also causes damage to biomolecules and metabolic and physiological dysregulation, which in turn causes imbalances in the immune function and inflammatory response of animals. The TLR signaling pathway is a signaling pathway closely related to anti-inflammatory immune mechanisms, and it plays an important role in the development of tissue inflammation [52]. TLR2 induces a myd88-dependent pathway that regulates other pro-inflammatory factors to indirectly induce a series of chain reactions that increase the expressions of pro-inflammatory factors [53,54]. The expression of myd88 was significantly increased in mice that were fed a high-fat diet, whereas myd88 expression was suppressed by the addition of 1600 mg/kg GML [55]. IL10 has significant effects on physiological processes, such as immune response and inflammation [56]. The expression of IL10 was significantly enhanced, and IL-1β tended to decrease in Danio rerio after the addition of 750 mg/kg of GML to the diet [26]. A study on piglets showed that the addition of 1000 mg/kg of α-GML resulted in a significant increase in the IL10 concentration [55]. The results of our experiment demonstrate that the addition of GML could reduce the expression of the pro-inflammatory factor IL-1β by decreasing the signaling of TLR4 and inhibiting the myd88-dependent pathway. The anti-inflammatory capacity of the organism was remarkably enhanced and could be sufficient to resist the inflammatory response brought about by oxidative stress, which was greatly related to the enhanced activity of the antioxidant enzyme system. The above experimental results demonstrate that GML could have a positive effect on reducing oxidative stress and improving immune function in grouper by regulating inflammation-related factors and increasing the activity of the antioxidant system.

When grouper are exposed to external stimuli, excess reactive oxygen species (ROS) are produced in the organism. ROS promotes lipid peroxidation in cell membranes, resulting in oxidative damage to the body and various diseases [57]. Similar to mammals, fish have both non-specific and specific immune functions, with various immune cells and factors in the blood, mucosal barriers, and tissues working together to protect against pathogens [58]. LZM is an important component of the specific immune function of fish. It lyses the bacterial cell wall by binding to the β-1,4 glycosidic bond between the hydrolysis sites N-Acetylmuramicacid (NAM) and *N*-Acetylglucosamine (NAG), resulting in the lysis of the bacteria [59]. Qian et al. found that GML (3000 mg/kg) significantly increased the survival rate and SOD activity of *Procambarus clarkii* infected with white spot syndrome virus (WSSV) [36]. An experiment in piglets showed that the addition of GML significantly reduced the MDA content and significantly increased the SOD activity of porcine epidemic diarrhea virus (PEDV)-infected piglets, while the change in CAT activity was not significant but tended to increase [56]. This study shows that the survival rate of grouper in a bacterial challenge test was significantly increased using GML (1800 and 2400 mg/kg). The LZM activity increased significantly after the bacterial challenge test and reached the maximum value in the G1800 and G2400 groups. At the same time, the bacterial challenge test had no significant effect on the MDA content when the GML addition level was greater than 1800 mg/kg. The possible mechanism for this is that GML can attach to the majority of G+ surfaces and a small amount of G-, leading to cell wall rupture and the efflux of intracellular material [59,60,61], significantly enhancing the ability of LZM to eliminate bacteria. The liver SOD activity increased significantly after the bacterial challenge test, with the highest activity in the G1800 and G2400 groups. At the same time, the bacterial challenge test had no significant effect on the liver MDA content when the GML addition level was greater than 1800 mg/kg. This indicates that the addition of GML resulted in a decrease in the accumulated ROS content of the organism and protected the body from oxidative damage.

The main metabolite of Firmicutes is butyric acid (BA) [62]. BA is a type of short-chain fatty acid that plays an active role in intestinal health. It is a major source of energy for intestinal epithelial cells and may promote intestinal health by increasing epithelial absorptive cells. The addition of GML, which significantly increased the abundance of thick-walled bacterial phylum in this experiment, could increase the accumulation of BA content and promote intestinal health [63]. In the association analysis between GML and intestinal flora, GML was correlated with *Bacillus*, *Psychrobacter*, *Acinetobacter*, *Acinetobacter*, *Stenotrophomonas*, and *Glutamicibacter*. *Bacillus* is one of the most widely used probiotics in aquaculture. *Bacillus*, as a probiotic, can improve antioxidant capacity, effectively scavenging free radicals and protecting the body from oxidative damage [64]. *Bacillus* can produce antimicrobial substances with broad-spectrum antimicrobial properties, inhibiting a wide range of pathogenic Gram-positive and Gram-negative bacteria (but more so for G+) and improving the immune system [65,66,67]. Cha et al. demonstrated that the addition of *Bacillus* to the diet significantly improved the immunity and disease resistance of *Paralichthysolivaceus* [68]. The above results have also been confirmed in *Oreochromis niloticus* [69] and *Litopenaeus vannamei* [70]. The results of this experiment show that GML promoted the growth of Bacillus, and the immune capacity, antioxidant capacity, and disease resistance were also improved. The carbohydrate utilization ability of carnivorous fish is much worse than that of terrestrial animals, and excessive carbohydrate content will lead to a decrease in fish resistance. Enteroglucagon has the function of promoting insulin secretion and inhibiting glucagon secretion, which can achieve hypoglycemic effect; however, it is easily degraded and inactivated by dipeptidyl peptidase-IV (DDP-IV) and cannot be effective. Yu et al. found that *Lysinibacillus* could inhibit DDP-IV activity in *Cyprinus carpio*, thus indirectly reducing blood glucose and promoting the health of the organism [71]. The application of *Lysinibacillus* to the culture of *Litopenaeus vannamei* significantly reduced the harmful nitrogen levels in the water and significantly increased the survival rate [72]. There is experimental evidence for the synthesis of silver nanoparticles from the extracellular filtrate of *Lysinibacillus sphaericus*, which have bactericidal activity against G- and G+ and are non-toxic to cells [73]. Studies have shown that *Glutamicibacter uratoxydans* KIBGE-IB41 produces chitinase, which hydrolyzes the cell wall of the fungus and enhances the ability of the organism to resist the fungus [74]. However, no study of *Glutamicibacter* in aquatic animals has been found, and this may also be considered a new finding. Studies in aquatic animals have shown that both *Acinetobacter* and *Psychrobacter* can secrete lipase [75], which is an important digestive enzyme for the body that hydrolyzes lipids and accelerates catabolism [76]. GML was significantly and positively correlated with *Acinetobacter* and *Psychrobacter* to increase the digestive activity of lipase and promote intestinal health in grouper.

## 5. Conclusions

In conclusion, the addition of GML significantly improved the antioxidant capacity, anti-inflammatory activity, and disease resistance of grouper, which effectively resisted oxidative damage caused by oxidative stress, enhanced the nutritional value of muscles, regulated intestinal flora, and increased immune capacity. This provides a theoretical basis for the application of GML in aquafeeds and greatly enhances the possibility of using GML in aquafeeds. Based on the above experimental results, the optimum level of GML in grouper feed is 1800 mg/kg.

## Figures and Tables

**Figure 1 metabolites-12-01089-f001:**
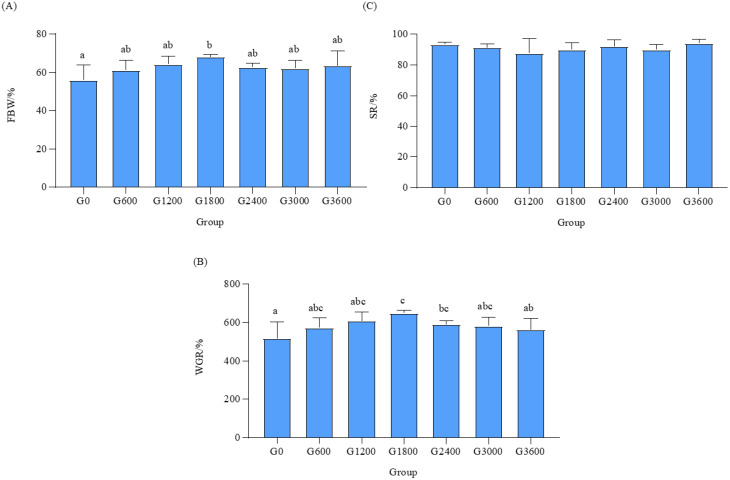
Effects of dietary GML levels on growth performance for juvenile grouper. Data represent means of three tanks in each group; error bar indicates S.D. Values with different letters are significantly different (*p* < 0.05). (**A**) FBW: weight body weight. (**B**) WGR: weight gain rate. (**C**) SR: survival rate.

**Figure 2 metabolites-12-01089-f002:**
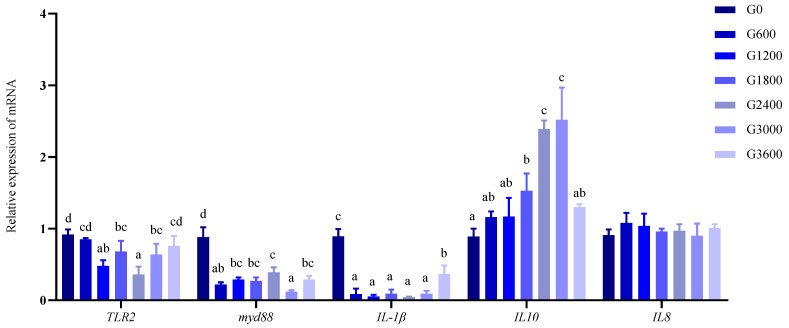
Effects of dietary GML levels on relative expressions of *TLR2*, *myd88*, *TL-1β*, *IL10*, and *IL8* in head kidney inflammatory factors genes in bacterial challenge test for juvenile grouper. Data represent means of three tanks in each group; error bar indicates S.D. Values with different letters are significantly different (*p* < 0.05). *TLR2*: toll-like receptor 2; *myd88*: myeloid differentiation primary response gene 88; *IL-1β*: interleukin 1β; *IL10*: interleukin 10; *IL8*: interleukin 8.

**Figure 3 metabolites-12-01089-f003:**
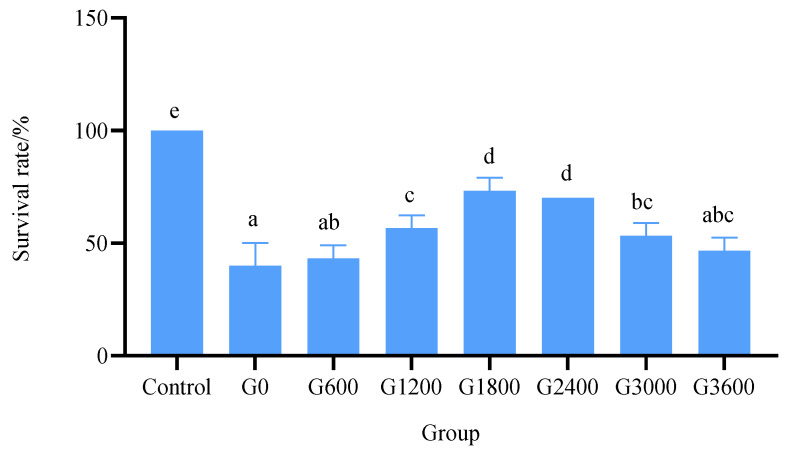
Effects of dietary GML levels on survival rate in bacterial challenge test for juvenile grouper. Data represent means of three tanks in each group; error bar indicates S.D. Values had different letters are significantly different (*p* < 0.05).

**Figure 4 metabolites-12-01089-f004:**
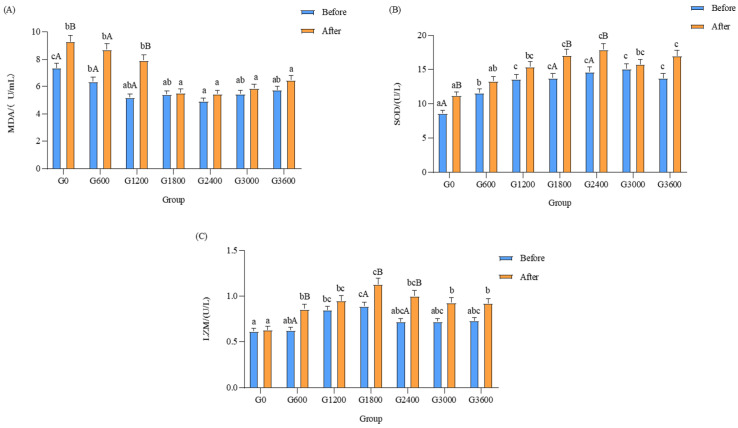
Effects of dietary GML levels on liver MDA (**A**), SOD (**B**) and LZM (**C**) activities in bacterial challenge test for juvenile grouper. Data represent means of three tanks in each group; error bar indicates S.D. Values with different letters are significantly different (*p* < 0.05). LZM: lysozyme. SOD: superoxide dismutase. MDA: malondialdehyde.

**Figure 5 metabolites-12-01089-f005:**
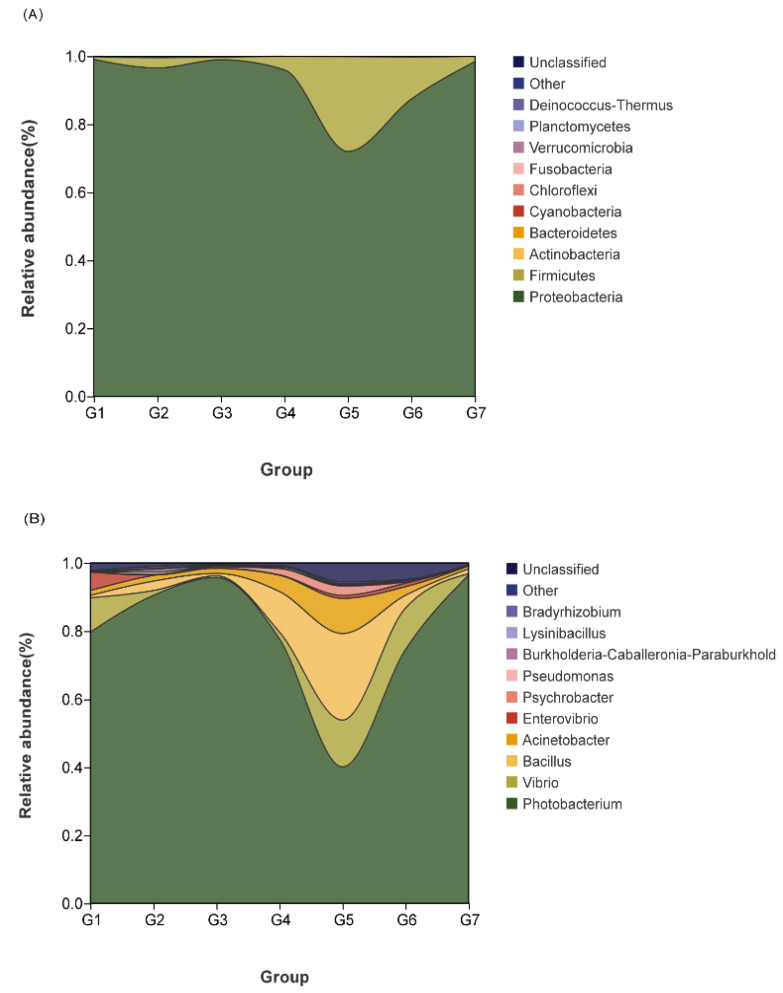
Effects of dietary GML levels on relative abundance of species at phylum (**A**) and genus (**B**) levels for juvenile grouper.

**Figure 6 metabolites-12-01089-f006:**
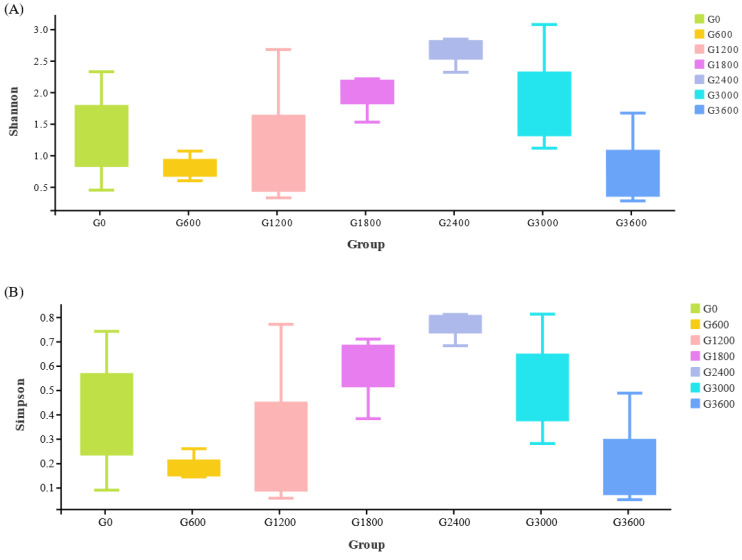
Effects of dietary GML levels on (**A**) Shannon and (**B**) Simpson indices for juvenile grouper.

**Figure 7 metabolites-12-01089-f007:**
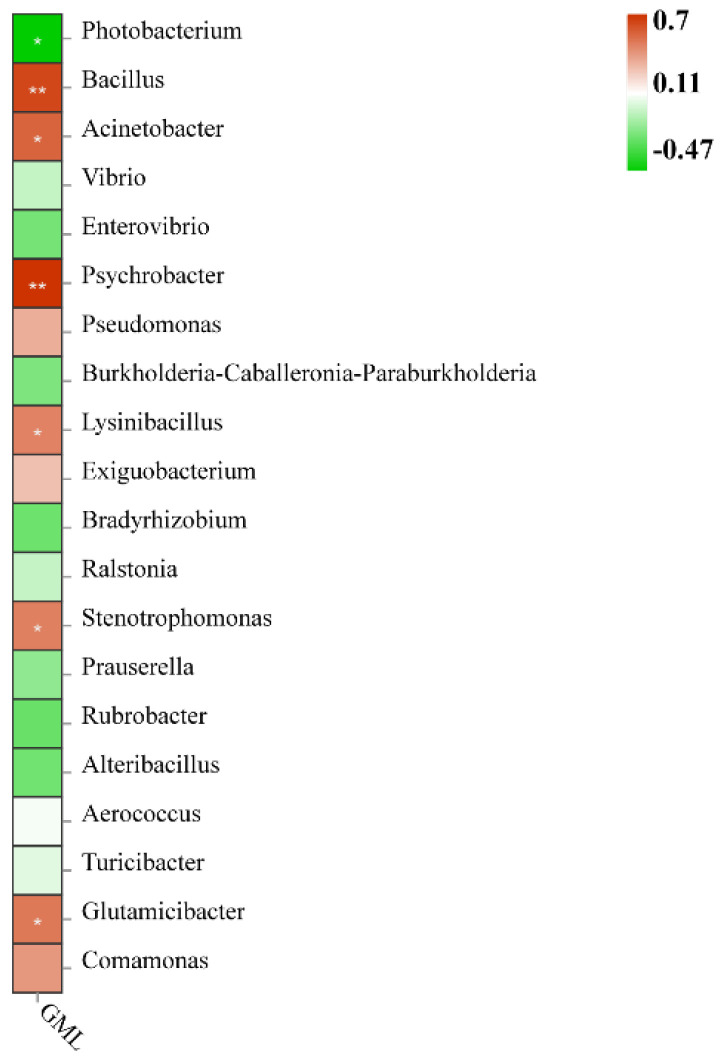
The association analysis of dietary GML and grouper intestinal flora genus. * represents significant difference, *p* < 0.05; ** represents highly significant difference, *p* < 0.01.

**Table 1 metabolites-12-01089-t001:** Ingredients and chemical compositions of experimental diets (% DM basis).

	Groups						
	G0	G600	G1200	G1800	G2400	G3000	G3600
Brown fish meal	30.00	30.00	30.00	30.00	30.00	30.00	30.00
American soybean meal	18.00	18.00	18.00	18.00	18.00	18.00	18.00
Corn gluten meal	9.50	9.50	9.50	9.50	9.50	9.50	9.50
Peanut meal	10.00	10.00	10.00	10.00	10.00	10.00	10.00
Chicken powder	6.00	6.00	6.00	6.00	6.00	6.00	6.00
Wheat flour	18.59	18.59	18.59	18.59	18.59	18.59	18.59
fish oil	2.00	2.00	2.00	2.00	2.00	2.00	2.00
Soybean oil	2.00	2.00	2.00	2.00	2.00	2.00	2.00
Soybean lecithin	1.00	1.00	1.00	1.00	1.00	1.00	1.00
Ca(H_2_PO_4_)_2_	1.00	1.00	1.00	1.00	1.00	1.00	1.00
Premix	1.00	1.00	1.00	1.00	1.00	1.00	1.00
Choline chloride	0.50	0.50	0.50	0.50	0.50	0.50	0.50
vitamin C	0.05	0.05	0.05	0.05	0.05	0.05	0.05
Microcrystalline cellulose	0.36	0.30	0.24	0.18	0.12	0.06	0.00
GML	0.00	0.06	0.12	0.18	0.24	0.30	0.36
Total	100.00	100.00	100.00	100.00	100.00	100.00	100.00
Nutrient levels							
Crude protein	49.74	49.19	49.65	49.66	49.19	49.32	49.40
Crude lipid	9.42	9.46	9.38	9.37	9.28	9.29	9.18

Premix (g/kg premix): vitamin A (675,000 IU), vitamin D_3_ (180,000 IU), vitamin E (6000 mg), vitamin K_3_ (1200 mg), vitamin B_1_ (900 mg), vitamin B_2_ (1350 mg), vitamin B_6_ (1050 mg), vitamin B_12_ (7.5 mg), calcium pantothenate (4500 mg), nicotinic acid (6750 mg), folic acid (375 mg), biotin (15 mg), vitamin C phosphate (42,860 mg), inositol (10,000 mg), FeSO_4_·H_2_O (64,286 mg), ZnSO_4_·H_2_O (26,283 mg), MnSO_4_·H_2_O (19,688 mg), Ca (IO_3_)_2_·H_2_O (128.6 mg), CoCl_2_·6H_2_O (323 mg), CuSO_4_·5H_2_O (2357 mg), Na_2_SeO_3_ (87.6 mg).

**Table 2 metabolites-12-01089-t002:** Primers pair sequences for real-time qPCR.

	Nucleotide Sequence (5′–3′)	Genbank Accession No.
*β* *-actin*	F: GATCTGGCATCACACCTTCT R: CATCTTCTCCCTGTTGGCTT	AY510710.2
*TLR2*	F: AGGGTTCAGAAGGGTTGCTAT R: CAGGAAGGAAGTCCCGTTTGT	HM357230.1
*myd* *88*	F: AGCTGGAGCAGACGGAGTG R: GAGGCTGAGAGCAAACTTGGTC	JF271883.1
*IL-1* *β*	F: AACCTCATCATCGCCACACA R: AGTTGCCTCACAACCGAACAC	XP_049460451.1
*IL-10*	F: ACACAGCGCTGCTAGACGAG R: GGGCAGCACCGTGTTCAGAT	KJ741852.1
*IL8*	F: GGCCGTCAGTGAAGGGAGTC R: TCAGAGTGGCAATGATCTCA	GU988706.1

*TLR2*: toll-like receptor 2; *myd88*: myeloid differentiation primary response gene 88; *IL-1β*: interleukin 1β; *IL10*: interleukin 10; *IL8*: interleukin 8.

**Table 3 metabolites-12-01089-t003:** Effects of dietary GML levels on antioxidant index of juvenile grouper.

	G0	G600	G1200	G1800	G2400	G3000	G3600
Liver							
MDA/nmol/mL	1.57 ± 0.26 ^d^	1.27 ± 0.21 ^bc^	1.08 ± 0.02 ^b^	0.86 ± 0.11 ^a^	1.28 ± 0.01 ^bc^	1.40 ± 0.01 ^cd^	1.14 ± 0.13 ^c^
SOD/U/mL	16.91 ± 0.46 ^a^	17.31 ± 0.31 ^a^	22.04 ± 0.42 ^c^	26.01 ± 0.73 ^d^	23.54 ± 1.14 ^cd^	21.38 ± 4.01 ^b^	18.64 ± 1.57 ^ab^
CAT/U/mL	8.29 ± 1.15	9.37 ± 0.11	10.19 ± 1.49	9.64 ± 0.79	9.31 ± 0.51	9.51 ± 0.97	8.84 ± 0.50
GSH-Px/U/mL	94.56 ± 12.43 ^a^	94.90 ± 13.80 ^a^	96.16 ± 5.01 ^ab^	113.53 ± 3.60 ^c^	108.86 ± 7.95 ^c^	106.96 ± 7.04 ^bc^	97.64 ± 8.87 ^ab^
Serum							
MDA/nmol/mL	1.66 ± 0.13 ^e^	1.73 ± 0.07 ^e^	1.12 ± 0.17 ^c^	0.79 ± 0.07 ^a^	1.00 ± 0.11 ^bc^	0.91 ± 0.11 ^ab^	1.36 ± 0.07 ^d^
SOD/U/mL	8.61 ± 0.56 ^a^	13.33 ± 1.94 ^b^	13.62 ± 2.12 ^b^	17.14 ± 0.67 ^c^	17.95 ± 1.71 ^c^	15.74 ± 2.34 ^bc^	17.01 ± 3.03 ^c^

Data are means ± SEM of three replicates (n = 3). Values within the same line with different superscripts are significantly different (*p* < 0.05). MDA: malondialdehyde; CAT: catalase; GSH-Px: glutathione catalase; SOD: superoxide dismutase.

**Table 4 metabolites-12-01089-t004:** Effects of dietary GML levels on fatty acid composition of juvenile grouper (%).

	G0	G600	G1200	G1800	G2400	G3000	G3600
C12:0	0.00 ± 0.00 ^a^	0.27 ± 0.01 ^b^	0.61 ± 0.01 ^c^	0.75 ± 0.04 ^d^	0.83 ± 0.01 ^e^	1.25 ± 0.12 ^f^	1.37 ± 0.03 ^g^
C14:0	2.56 ± 0.07 ^a^	2.77 ± 0.00 ^bc^	2.68 ± 0.08 ^ab^	2.70 ± 0.25 ^ab^	2.94 ± 0.01 ^c^	2.94 ± 0.01 ^c^	2.78 ± 0.02 ^bc^
C16:0	19.87 ± 0.16 ^b^	20.41 ± 0.09 ^c^	19.73 ± 0.08 ^b^	19.11 ± 0.05 ^a^	20.48 ± 0.32 ^c^	20.36 ± 0.21 ^c^	19.81 ± 0.07 ^b^
C16:1n7	2.85 ± 0.03 ^a^	3.41 ± 0.02 ^f^	3.04 ± 0.02 ^c^	2.95 ± 0.03 ^b^	3.29 ± 0.02 ^e^	3.24 ± 0.02 ^d^	3.03 ± 0.03 ^c^
C17:0	0.35 ± 0.01 ^b^	0.35 ± 0.01 ^b^	0.39 ± 0.01 ^c^	0.41 ± 0.01 ^c^	0.40 ± 0.01 ^c^	0.41 ± 0.02 ^c^	0.29 ± 0.03 ^a^
C18:0	8.18 ± 0.13 ^f^	6.89 ± 0.03 ^a^	7.77 ± 0.09 ^e^	7.45 ± 0.03 ^c^	7.25 ± 0.04 ^b^	7.74 ± 0.04 ^e^	7.62 ± 0.01 ^d^
C18:1n9c	17.84 ± 0.28 ^a^	19.94 ± 0.14 ^e^	18.91 ± 0.03 ^d^	18.63 ± 0.06 ^bc^	19.11 ± 0.13 ^d^	18.62 ± 0.26 ^bc^	18.32 ± 0.06 ^b^
C18:2n6c	24.09 ± 0.08 ^c^	25.25 ± 0.10 ^e^	24.98 ± 0.05 ^d^	25.30 ± 0.08 ^e^	23.91 ± 0.22 ^bc^	23.14 ± 0.14 ^a^	23.79 ± 0.21 ^b^
C18:3n3	0.00 ± 0.00 ^a^	3.98 ± 0.14 ^c^	3.54 ± 0.04 ^b^	3.55 ± 0.03 ^b^	0.00 ± 0.00 ^a^	0.00 ± 0.00 ^a^	0.00 ± 0.00 ^a^
C20:0	0.45 ± 0.01 ^e^	0.31 ± 0.00 ^abc^	0.33 ± 0.02 ^bcd^	0.30 ± 0.01 ^ab^	0.34 ± 0.02 ^cd^	0.34 ± 0.01 ^d^	0.29 ± 0.02 ^a^
C20:1	3.32 ± 0.04 ^b^	0.00 ± 0.00 ^a^	0.00 ± 0.00 ^a^	0.00 ± 0.00 ^a^	3.56 ± 0.02 ^d^	3.50 ± 0.06 ^d^	3.42 ± 0.07 ^c^
C20:2	3.32 ± 0.04 ^b^	0.00 ± 0.00 ^c^	0.00 ± 0.00 ^d^	0.00 ± 0.00 ^e^	3.56 ± 0.02 ^a^	3.50 ± 0.06 ^bc^	3.42 ± 0.07 ^d^
C20:3n6	0.34 ± 0.01 ^d^	0.23 ± 0.02 ^b^	0.28 ± 0.03 ^c^	0.26 ± 0.01 ^c^	0.28 ± 0.01 ^c^	0.38 ± 0.01 ^e^	0.20 ± 0.01 ^a^
C20:4n6	1.31 ± 0.04 ^e^	1.02 ± 0.02 ^ab^	1.04 ± 0.04 ^abc^	1.09 ± 0.03 ^e^	0.99 ± 0.06 ^a^	1.10 ± 0.01 ^c^	1.17 ± 0.05 ^d^
C20:5n3	6.12 ± 0.10 ^d^	5.80 ± 0.02 ^b^	5.58 ± 0.04 ^a^	5.96 ± 0.07 ^bc^	5.90 ± 0.04 ^bc^	5.89 ± 0.04 ^bc^	6.11 ± 0.06 ^d^
C22:0	0.45 ± 0.00 ^d^	0.30 ± 0.02 ^b^	0.38 ± 0.03 ^c^	0.32 ± 0.01 ^c^	0.31 ± 0.02 ^b^	0.34 ± 0.02 ^b^	0.24 ± 0.01 ^a^
C22:1n9	0.36 ± 0.01 ^bc^	0.39 ± 0.00 ^c^	0.35 ± 0.03 ^b^	0.37 ± 0.03 ^bc^	0.31 ± 0.01 ^a^	0.34 ± 0.02 ^ab^	0.35 ± 0.02 ^b^
C22:6n3	8.92 ± 0.14 ^c^	7.50 ± 0.13 ^a^	8.39 ± 0.05 ^b^	9.23 ± 0.08 ^d^	8.86 ± 0.02 ^c^	8.92 ± 0.14 ^c^	9.62 ± 0.20 ^e^
C24:1n9	0.87 ± 0.07 ^d^	0.53 ± 0.03 ^a^	1.08 ± 0.03 ^e^	0.59 ± 0.03 ^ab^	0.62 ± 0.02 ^bc^	0.68 ± 0.06 ^c^	0.66 ± 0.06 ^bc^
∑SFA	31.86 ± 0.21 ^b^	31.29 ± 0.08 ^a^	31.88 ± 0.11 ^b^	31.04 ± 0.16 ^a^	32.56 ± 0.31 ^c^	33.38 ± 0.18 ^d^	32.41 ± 0.08 ^c^
∑n-3	15.04 ± 0.12 ^b^	17.28 ± 0.02 ^d^	17.51 ± 0.09 ^d^	18.74 ± 0.16 ^e^	14.75 ± 0.05 ^a^	14.81 ± 0.19 ^ab^	15.73 ± 0.24 ^c^
∑n-6	25.74 ± 0.12 ^c^	26.49 ± 0.07 ^de^	26.30 ± 0.08 ^d^	26.65 ± 0.06 ^e^	25.19 ± 0.27 ^b^	24.62 ± 0.13 ^a^	25.15 ± 0.19 ^b^
∑n-3/∑n-6	0.58 ± 0.00 ^a^	0.65 ± 0.00 ^d^	0.67 ± 0.00 ^d^	0.70 ± 0.01 ^e^	0.59 ± 0.01 ^a^	0.60 ± 0.01 ^b^	0.63 ± 0.01 ^c^

SFA: saturated fatty acid; n-3: n-3 polyunsaturated fatty acids; n-6: n-6 polyunsaturated fatty acids. Values within the same line with different superscripts are significantly different (*p* < 0.05).

## Data Availability

The data presented in this study are available on request from the corresponding author. The data are not publicly available due to [the original data is only stored on private computers and not published to public databases].

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
