# Peer review of "Effects of Plant-Derived Glycerol Monolaurate (GML) Additive on the Antioxidant Capacity, Anti-Inflammatory Ability, Muscle Nutritional Value, and Intestinal Flora of Hybrid Grouper (Epinephelus fuscoguttatus♀ × Epinephelus lanceolatus♂)"

_metabolites, 2022, doi:10.3390/metabo12111089_

Round 1
Reviewer 1 Report
The article by Xuehe Li et al., 2022 titled “The effect of the plant-derived additives glycerol monolaurate (GML) on the antioxidant capacity, anti-inflammatory ability, muscle nutritional value and intestinal flora of hybrid grouper (Epinephelus fuscoguttatus x Epinephelus lanceolatus” was carefully reviewed and suggested the following changes.
1. Introduction is short, make it more strengthened according to the title of the study.
2. Give reference from which this idea was conceived.
3. The authors only analyze IL-10 and myd88 genes, do you think these two genes are sufficient to determine anti-inflammatory and inflammatory response, justify with solid references
4. The authors analyzed the antioxidant indices from liver and serum but from liver 4 antioxidant assays were measured, and two from serum, why? Justify it.
5. Make graphs using Prism GraphPad software.
6. Write abbreviations separately.
7. How your studies make difference from already available literature?
8. Do you think your study could replace antibiotics usage in feed?
9. There are some minor mistakes in English, must be corrected.
Author Response
Dear Reviewer,
Please see the attachment for response comments.
Best wishes,
Xuehe Li

Reviewer 2 Report
Comments of Reviewer
The examined research on “The effect of the plant-derived additives glycerol monolaurate (GML) on the antioxidant capacity, anti-inflammatory ability, muscle nutritional value and intestinal flora of hybrid grouper (Epinephelus fuscoguttatus × Epinephelus lanceolatus)” presents an interesting piece of work. The topic is novel where plant derivatives are used for exercising the health and muscle quality. However, I feel further improvement can be made and my personal observations are appended below.
Abstract
The focus of the work is on glycerol monolaurate (GML). I suggest adding a line about its novelty at the start of the abstract. Change the word “aquatic feed” to “aquafeed”.
Introduction
1. The last paragraph needs rewriting. Please exercise the research gap and objectives to support the work in a clear way.
Methods
2. The present study was approved by the Guangdong Ocean University ethics review board. Elaborate the approval no. and date if available.
3. Line 97: “evenly broken” should be replaced with alternate word., e.g. homogenized
4. Line 119: 4000 r/min should be replaced as rpm
5. The statistical method is missing. Include.
Results
6. 167: Italicize scientific name
Discussion
This section need to be carefully revised as several flaws are noticed. Some sections are repeated unnecessarily. Scavenging effort has been repeated in few instances. Carefully, correct it.
I hope the research piece can be considered provided the suggested changes are incorporated.
Author Response

(The authors gave the same response as above.)

Round 2
Reviewer 1 Report
The article was revised as per suggestion.